# Unraveling the Pancreatic Anlagen: Validating a Manual Dissection Protocol with Immunohistochemical Staining for Pancreatic Polypeptide in a Human Cadaver Study

**DOI:** 10.3390/biomedicines13061318

**Published:** 2025-05-28

**Authors:** Athanasios Alvanos, Elisa Schubert, Karsten Winter, Hanno Steinke

**Affiliations:** Institute of Anatomy, Faculty of Medicine, University of Leipzig, Liebigstraße 13, 04103 Leipzig, Germany; elisa.schubert@medizin.uni-leipzig.de (E.S.); kwinter@rz.uni-leipzig.de (K.W.); hanno.steinke@medizin.uni-leipzig.de (H.S.)

**Keywords:** pancreatic anlagen, morphogenetic units, embryological fusion plane, connective tissue, pancreatic polypeptide, pancreatic cancer

## Abstract

**Background**: The pancreas develops from two independent buds that fuse to form the adult organ. Ontogeny has largely been neglected in pancreatic surgery. This study aims to demonstrate that the adult pancreas can still be divided into morphogenetic units based on its embryological compartments and connective tissue borders for potential therapeutic purposes. **Methods**: Ten donor bodies (four female, six male, aged 73–101 years) were used. Manual dissection, guided by the common bile duct to locate the embryological fusion plane, was performed to divide the pancreatic tissue. Immunohistochemical staining for pancreatic polypeptide differentiated the pancreatic tissue by embryological origin and was used to quantify dissection accuracy. **Results**: Landmark-guided dissection successfully separated the pancreas along a connective tissue plane in seven cases. The resulting compartments were distinctly divided along the dissection plane into an area rich in pancreatic polypeptide and an area with low accumulation. Two cases showed deviations from the dissection plane at the histological level. One case contained tumor tissue, interfering with the utilization of landmarks. **Conclusions**: Landmark-guided dissection of the pancreas based on its embryological fusion plane allows for reliable separation into morphogenetic compartments. Immunohistochemical staining for pancreatic polypeptide effectively differentiates tissue origins. This approach may enable more precise, differentiated pancreatic resections and tailored treatments, with potential for refinement in routine surgical practice. Approaching the pancreatic tissue with regard to its ontogenetic origin and its clearly distinguishable compartments might even enable tailored treatment beyond refined surgical procedures.

## 1. Introduction

In a recent epidemiologic cancer study by Siegel et al., patients with pancreatic cancer had a 5-year survival rate of only approximately 12% [1]. More than 90% of pancreatic cancers arise from exocrine cells, and ductal adenocarcinoma constitutes the vast majority of exocrine pancreatic cancers [2,3]. The main location is the head of the pancreas, where more than 70% of the tumors are found [4]. A surgical approach remains the only curative option if the cancer is resectable [5]. However, ontogenetic considerations, potentially revolutionary in the surgical treatment of cancer, are yet to be thoroughly investigated in pancreatic cancer.

A surgical approach for cervical cancer, considering an ontogenetic approach to isolate the specific compartments for the spread of cancer cells, has been pioneered by Höckel et al. [6,7].

Interestingly, these compartments encompass tissue of the same embryological origin and hence, were termed “morphogenetic units” [6,8,9]. Resection of a compartment affected by cancer reduces the risk of locoregional recurrence, as long as the spread of the malignancy has not breached the developmentally defined boundaries formed by the connective tissue [6,7,8,9,10,11]. This approach contrasts with resection, which is based on the distance from the malignancy.

The pancreas seems to be the ideal organ for a comparable surgical approach, based on the premise of ontogenesis. While it appears as a homogeneous organ, it derives from two distinct anlagen, namely the ventral and dorsal pancreatic buds, that develop independently [12]. It appears to be an oversight to conceptualize the pancreas as a single unit stemming from a single anlage rather than two independent, yet fused, morphogenetic units for potential therapeutic or surgical approaches.

As previously demonstrated, it is possible to separate anlagen along an embryological fusion plane that remains visible in the adult organ [13]. In the current study, manual dissection along the fusion plane was validated by immunohistochemical staining for pancreatic polypeptide (PP), a valid marker of ventral pancreatic bud-derived pancreatic tissue [14,15,16,17]. It was demonstrated in human donors that pancreatic anlagen can be reliably separated using anatomical landmarks and can be distinguished immunohistochemically. We aimed to provide quantifiable evidence for the viability of this dissection approach in terms of consistent results, which are extremely important in surgical oncology.

## 2. Materials and Methods

### 2.1. Demographic Information, Logistics, and Ethics Regarding the Body Donors

For this study, tissues from 10 humans (four females and six males) were analyzed immediately post-mortem. The ages ranged from 73 to 101 years, with an average age of 88.1 ± 8.16 years and a median age of 87.5 years (Appendix A). Additionally, one female donor provided tissue for demonstrating the manual dissection technique in the figures, whereas another provided tissue for the Appendix A. These donors were not included in the statistical analysis or examinations beyond manual dissection.

The donors included in this study showed no signs of previous abdominal surgery, such as abdominal scarring or extensive adhesions. No donor displayed advanced malignancies of the abdominal organs or clinically significant pancreatic pathologies, as indicated by the death certificate documentation.

The bodies were cooled post-mortem at 3 °C until the dissection took place. After manual dissection and tissue preservation, the bodies were cremated or prepared for educational or scientific purposes at the Institute of Anatomy, depending on the declared will of the donor.

The donors provided informed consent for the use of their bodies for scientific and educational purposes at the Institute of Anatomy at the University of Leipzig in accordance with the Saxonian Death and Funeral Act of 1994. The signed consent documents can be provided by the corresponding author upon reasonable request. Approval from the Institutional Review Board of the University of Leipzig was granted for the scientific use of donor bodies on 1 March 2022 (protocol number 129/21-ek).

### 2.2. Manual Dissection

Manual dissection was performed by the same two operators. One operator is an anatomist with extensive experience in anatomical dissection, while the other operator has completed his senior residency in the surgical department at the University Hospital of Leipzig, where pancreatic surgery is frequently performed.

The Cattell–Braasch maneuver, which has been previously presented, was applied to fully expose the pancreas [13]. After ligatures were placed at both ends of the duodenum, the liver hilum, and the splenic hilum, the pancreas was extracted, along with the adjacent duodenum (Figure 1). The common bile duct was preserved as an important landmark for further dissection. The topographic anatomy of the extracted specimen and the adjacent structures is presented in Appendix A.

Dissection was continued in a craniocaudal direction along the ventral surface of the common bile duct, following the connective tissue of the assumed embryological fusion plane between the pancreatic compartments (Appendix A). The embryological fusion plane was breached by the main pancreatic duct, which led from the tissue derived from the dorsal pancreatic bud to the portion formed by the ventral pancreatic bud (Appendix A). To isolate the compartments further, the duct was transected (Appendix A). To validate this finding, the pancreatic duct was probed in the directions of the duodenum and pancreatic tail. The compartments were separated along the assumed embryological fusion plane, further exposing the ventral surface of the tissue derived from the ventral pancreatic bud (Figure 2). Subtotal separation of the pancreatic compartments along the embryologic fusion plane, including transection of the main pancreatic duct, is presented in Appendix A.

To create the specimen for histological examination, the compartments were not completely separated. Instead, after reaching the stage shown in Figure 2, a sagittal incision was made on the right side of the common bile duct, dividing the pancreatic head (Appendix A). The resulting cross-sectional view resembles a V-shape, with the ventral bud forming tissue of the left leg of the “V” and the dorsal bud forming the tissue of the right leg of the “V”, corresponding to the dorsal and ventral side of the specimen (Figure 3A). This process and the resulting cross-sectional view of the pancreatic head is demonstrated in Appendix A.

After making a further sagittal incision approximately 2 cm to the left, the final specimen, which would be embedded for further immunohistochemical examination, was created (Figure 3B). The lateral view of this cross-sectional slice of the pancreatic head is demonstrated in Appendix A, and topography with regard to the embryologic origin is explained.

As the tissue formed by the dorsal pancreatic bud was considered a negative control for the subsequent immunohistochemical staining for PP, a pancreatic cross-section 1 cm from the far end of the pancreatic tail was excised (Figure 3A).

The secured tissues were preserved in 4% paraformaldehyde (PFA) and stored at room temperature until further histological preparation and analysis.

### 2.3. Preparation of Histological Sections and Immunohistochemical Staining for Pancreatic Polypeptide (PP)

The specimens were embedded in paraffin and cut into 10 μm slices. After deparaffinization, the sections were subjected to hematoxylin and eosin and immunohistochemical staining. Immunohistochemical staining was initiated by unmasking antigens using an antigen retriever (2100 Retriever, Aptum Biologics Ltd., Southampton, UK) and a citrate buffer (pH = 6.0). The sections were washed thrice with 0.3% Triton X-100 in phosphate-buffered saline (PBST). They were first incubated in normal goat serum for 30 min and then incubated with an anti-PP antibody (polyclonal; Merck Cat. #AB939-I, Rahway, NJ, USA; diluted 1:2000 in 0.5% bovine serum albumin in PBST) in a humidified chamber overnight at 4 °C. After washing three times with PBST, the cells were incubated with a secondary anti-rabbit antibody (polyclonal; Invitrogen Cat. #A-11036, Waltham, MA, USA; diluted 1:500 in PBST) for 60 min, followed by washing with phosphate-buffered saline (PBS) and nuclear staining with 4′,6-diamidino-2-phenylindole (DAPI; Sigma Aldrich, St. Louis, MO, USA) for 5 min. The sections were then washed with PBS and treated with TrueBlack^®^ (Biotum, Hayward, CA, USA) to reduce autofluorescence. After washing, the sections were covered with a fluorescence mounting medium (Dako) and stored for further analysis.

### 2.4. Digitization and Analysis of Immunostained Tissue Sections

After immunohistochemical staining, histological sections of the pancreatic head and negative controls were digitized using a Zeiss AxioScan 7 digital slide scanner (Carl Zeiss Microscopy GmbH, Jena, Germany) at 5× magnification. For digitization, the blue channel (DAPI filter) was used to capture the DAPI signals, and the red channel (AF568 filter) was used for the PP signals. Individual channel images and combined RGB images were exported as raster images with a pixel size of 2.752 µm using ZEN slide scanner software (Version 3.7, Carl Zeiss Microscopy GmbH, Jena, Germany).

RGB images were imported into GIMP (Version 2.10.34; The GIMP team, http://www.gimp.org) (Appendix A). The contrast was adjusted, and the tissue regions of interest (ROI) were manually delineated, depending on the accumulation of PP (green for PP-rich, white for PP-poor; Appendix A). Large confluent areas of fatty tissue that did not contain pancreatic tissue were excluded. However, vessels or scattered islands of nonpancreatic tissue within the ROI were not selectively excluded. The original channel images and images containing the respective ROIs were imported into Mathematica for further processing (Version 14; Wolfram Research Inc., Champaign, IL, USA).

To detect pancreatic tissue in the DAPI channel, a minimum error global thresholding method [18] was applied, followed by morphological closing and removal of small specks. PP signals were detected using another global thresholding algorithm [19], followed by morphological closing (Appendix A). Manually drawn ROIs were extracted and converted into area masks. These masks were multiplied by their respective DAPI- and PP-signal areas. The areas covered by DAPI and PP pixels within each mask were counted, and the PP coverage (PP area per DAPI area in percentage) was computed. The PP coverage of the respective areas was averaged for all sections.

Next, the tissue edges along the dissection line were analyzed to determine the accuracy of surgical separation into PP-poor and PP-rich areas (Appendix A). The starting point of the dissection line is marked in pink, and the end point of the dissection line is marked in yellow (Appendix A). The curve shapes of both tissue edges were extracted from the marked image by morphological thinning of the respective white and green line segments. In the second step, normal vectors were calculated for each pixel of the resulting curves, and virtual incisions with a length of 2 mm were generated (Appendix A) and transferred onto both tissue edges (Appendix A).

For each incision, the underlying pixels from the tissue and PP-signal detection were extracted and depicted in a graph, layer-by-layer (Figure 4A). Each compartment was depicted on one-half of the graph. The colored bars in the center of the graph between the opposing sides indicate tissue designation during the dissection and manual delineation processes, based on immunohistochemical staining (Figure 4A).

Subsequently, an algorithmic method of tissue designation was added. Red and blue pixels were sorted and counted for each layer (Figure 4B). The PP-signal strength was analyzed to determine the tissue affiliation. The number of red pixels per layer was converted to a list and smoothed with a median filter (window size of 50), followed by closing of gaps smaller than 50 layers wide (Figure 4C, orange line). The data were then thresholded using a minimum error global thresholding method [18] to assign the area to the PP-rich area, as indicated by an increased local number of red pixels (Figure 4C, cyan line). This algorithmic assignment of areas is represented by a third color-coded bar (Figure 4D).

Using these data, the accuracy of the manual dissection of the pancreatic head into two distinct areas was quantified. The concordance ratio was calculated for the macroscopic designation (central bars) with the delineation of the tissue edges based on immunohistochemical appearance (second bars) and with the algorithmic assignment of tissue edges to the distinctive areas (third bars). The conformance of the algorithmic assignment and manual delineation were then compared.

## 3. Results

### 3.1. Separation of the Pancreatic Buds and Identification of Developmentally Defined Boundaries

Manual dissection led to division of the pancreas along the denser connective tissue in front of the common bile duct in 9 of the 10 study cases at a macroscopic level (Figure 2), failing only in donor 8. The connective tissue presented a plane that was breached by the main pancreatic duct (Appendix A). This breach appeared consistently, except in donors 8 and 9, identifying the main pancreatic duct as a reliable landmark.

In the case of donor 8, tumor distorted the tissue to an extent, which rendered the landmark insufficient for a macroscopically satisfactory result and prevented the identification of the connective tissue for sufficient separation of the compartments along a guiding plane (Appendix A).

Given the consistency of the connective tissue and its trajectory, it was identified as the embryological fusion plane. On the premise that the fusion plane marks the adhesive zone between the compartments derived from the ventral and dorsal pancreatic buds, an effort was made to label the connective tissue that was considered to form the developmental boundaries of the pancreatic compartments (Figure 5). The “V”-shaped cross-section of the histological specimen allows for a lateral view of the assumed topographic anatomy of these boundaries (Figure 6).

### 3.2. Comparing PP-Coverage in the Respective Areas

The calculation of PP coverage was based on the manual designation of tissue areas based on the amount of visible PP (Appendix A). The PP coverage in the negative control was 0.14% ± 0.09. In the head sections, the coverage in the PP-rich area was 2.58% ± 1.69, while the PP-poor area showed a coverage of 0.12% ± 0.06 (Figure 7).

To evaluate whether there were significant differences in PP coverage, the Shapiro–Wilk test was used to check for data normality, which was not the case. Group comparisons were then performed using the Kruskal–Wallis test with post hoc Bonferroni correction. The significance level for all tests was set at *p* < 0.05. The coverage in the PP-rich area differed significantly from that in the PP-poor area, as did the coverage in the negative control (*p* ≤ 0.001). More importantly, the coverage in the pancreatic tail did not differ significantly from that in the PP-poor area of the pancreatic head (*p* = 0.660).

### 3.3. Juxtaposition of the Macroscopic Tissue Designation and the PP-Immunostaining

The alignment of macroscopic tissue designation and PP coverage (ventral pancreatic bud = PP-rich; dorsal pancreatic bud = PP-poor) was found in 100% of the negative controls. As expected, tissue extracted from the pancreatic tail, which was assigned to the dorsal pancreatic bud, was consistently PP-poor (Figure 8).

For the pancreatic head, the macroscopic tissue designation was validated by immunostaining in 7 of 10 cases. Here, immunostaining revealed a clean separation into PP-poor and PP-rich compartments (Figure 9). In two of the remaining cases (donors 2 and 6), the entry point for the dissection was poorly chosen, thereby damaging the integrity of one of the compartments (Figure 10 and Appendix A), but the dissection eventually managed to return to the correct plane of the connective tissue. In the case distorted by tumor tissue (donor 8; Appendix A), the entry point was also chosen incorrectly. Thus, without the necessary landmarks, the process could not be salvaged; thus, separation was unsuccessful. The immunohistochemically stained histological sections, before digital adjustment of contrast for Figure 8, Figure 9 and Figure 10, can be viewed in the Appendix A (Appendix A, respectively).

In two cases (donors 4 and 10), the separated compartments were realigned during the embedding process, making it difficult to identify the artificial gap in the histological sections. The overlay of the macroscopic images verified that the dissection plane sufficiently divided the areas.

All concordance ratios between the macroscopic tissue designation and the manual assignment based on immunostaining, as well as the concordance of the algorithmic assignment with the macroscopic tissue designation, were calculated and summarized (Figure 11).

## 4. Discussion and Conclusions

It has previously been shown that it is possible to reverse the fusion of the pancreatic anlagen within the pancreatic head via manual dissection [13]. In the current study, a detailed procedure was demonstrated to reliably separate the pancreatic anlagen, and the procedure was validated by immunohistochemical staining for PP.

Anatomical studies have repeatedly shown how PP can be used as a selective marker for tissue derived from the ventral pancreatic bud [14,15,16,17]. However, the only group that provided a detailed approach for tissue separation in a cadaver study was Sakamoto et al. [20]. However, although immunohistochemical quality control was claimed in the study, no evidence was provided. In clinical cases involving patients, selective segmentectomy along the embryological fusion plane has been performed and published in sporadic case reports [21,22,23,24,25,26,27,28]. These case reports do not immunohistochemically validate the selectivity of resection, except for the work of Talbot et al., which confirmed through immunostaining for PP that the resected tissue did indeed belong to the dorsal pancreatic bud [25]. To date, the combination of these methods has been insufficient.

In this study, anatomical landmarks were used to separate the derivatives of the dorsal and ventral pancreatic buds along the embryological fusion plane. This fusion plane presents itself as a plane of dense connective tissue. A very consistent anatomy was encountered, allowing for a standardized procedure (Figure 1, Figure 2 and Figure 3 and Appendix A). However, the legitimacy of this procedure can only be confirmed by immunohistochemical analysis. Immunostaining for PP is a reliable determinant of tissue affiliation to either pancreatic bud. A highly significant difference in PP coverage on either side of the dissection plane was observed. A significantly higher amount of PP was observed in the dorsally situated compartment derived from the ventral pancreatic bud (Figure 7). This is consistent with previous research [14,15,16,17]. In addition, the negative control from the pancreatic tail could be assigned to the tissue derived from the dorsal pancreatic bud, judging by PP coverage, which was the expected outcome (Figure 7 and Figure 8).

A successful procedure is therefore defined as surgical separation along the embryological fusion plane, which results in a satisfactory overlap of the macroscopic designation of the tissue to the specific pancreatic bud, with the resulting accumulation of PP, as shown in Figure 9. This grade of clean separation was observed in 7 of 10 cases (Figure 11). Choosing the correct entry point for dissection is essential; otherwise, the integrity of the compartments will be compromised, as seen in two cases (Figure 10 and Appendix A). Notably, in a case involving tumorous tissue, the tissue was distorted in a way that rendered the use of landmarks insufficient and prevented determining the correct entry point for dissection (Appendix A). The main pancreatic duct, which is the only major structure that breaches the embryological fusion plane, is also a reliable indicator of the correct dissection plane (Appendix A), but it could not be used in the tumorous sample. To achieve more objective tissue designation, an algorithmic assignment of the tissue edges was implemented. The concordance ratio between the manual and algorithmic assignments was consistently high, although the algorithmic assignment appeared to understate the results by falsely assigning tissue to PP-poor areas, as it did not distinguish between pancreatic tissue and vessels, which are inherently PP-poor. In addition, the algorithmic assignment was based on virtual tissue incisions that were only 2 mm deep rather than on the entire tissue. The results presented above promise clinical significance for several reasons, as discussed below.

First, the ability to revert embryological rotation and adhesion along the embryological fusion plane reestablishes independent pancreatic units. We have repeatedly advocated the view of Höckel, who supports the theory that tumor cells first spread within the “morphogenetic unit” in which they emerge [6,8,9]. This unit encompasses tissues of a common embryological origin [6,8,9]. An incomplete resection of a compartment might lead to a resurgence of cancer cells, be it due to cancer cells that had already spread within the compartment; lesions that are present within the residual compartment, even if the resection margins appear to be unaffected; or other unknown properties of these compartments [11]. Sufficient resection would remove all tissue based on the ontogenetic relationship instead of indiscriminate resection based on the mere vicinity to the tumor [6,8,9,11]. Höckel’s approach has led to major advances in the surgical treatment of cervical cancer, especially regarding locoregional tumor control, by applying the total mesometrial resection method [7,10].

For pancreatic head cancer, this would suggest a tailored approach, depending on whether it originates from the dorsal or ventral pancreatic bud. In both cases, radical resection of the compartment in which the cancer can potentially spread is required. The integrity of the removed compartment could then be verified by immunostaining, which proved to be a reliable marker. However, current Whipple-like procedures do not make these distinctions; Whipple-like procedures remove the head of the pancreas but ultimately fail to reliably remove either anlage. Given this approach to cancer surgery and reflecting on Höckel’s approach, it is not surprising that even in cases of resections where the margins are deemed tumor-free, pancreatic cancer recurrence rates of 60–100% are reported [29,30,31]. Instead, with selective but thorough resection of one compartment, the eradication of malignant cells might be more effective.

To enable a tailored approach for pancreatic cancer, developmentally defined boundaries for pancreatic compartments must be identified. The current study builds on previous work [13] and labels candidates for the fascia of Treitz, the fascia of Toldt, the mentioned embryological fusion plane, and the unnamed prepancreatic layer (Figure 5 and Figure 6). These connective tissues might serve as said boundaries.

Second, it is necessary to study pancreatic anlagen, as well as the cancer that originates within, independently. The ability to reliably distinguish the two compartments anatomically and immunohistochemically was demonstrated (Figure 7 and Figure 11), which is a prerequisite for this task. Currently, research focusing on the epidemiology, therapy, and prognosis of pancreatic cancer with regard to embryological anlagen is scarce. Nevertheless, a few studies have shown substantial differences between these locations. Foremost, the part of the pancreas that originates from the ventral bud appears to be the predominant location for carcinogenesis, judging by the number of cases [16,17,32]. Moreover, there is a consistent disparity in lymphatic metastatic behavior and metastatic behavior along the nerve plexus [16,17,32]. Lymph node stations 8 and 12, located at the common hepatic artery and hepatoduodenal ligament, are exclusively affected by the drainage of cancer originating from the derivative of the dorsal pancreatic bud [17]. In contrast, only cancers originating from the derivative of the ventral pancreatic bud drained into lymph node station 14 towards the superior mesenteric artery. Metastasis along the nerve plexus follows the same pattern [16]. Malignancies of the dorsal and ventral pancreatic buds spread towards the nerve tissue surrounding the common hepatic artery and superior mesenteric artery, respectively. Recently, a study has shown that the timing of metastasis differs between the two buds [4]. Gao’s group described how cancer of the tissue of the dorsal pancreatic bud propagates much quicker than its counterpart in the ventral pancreatic bud and is more likely to show microvascular invasion [4]. This aligns with another study, which has also shown better survival for patients with cancer derived from the ventral pancreatic bud [32], in which case metastases appear later.

These findings, combined with the knowledge of a separate blood supply [33,34,35] and separate nerve supply [36] of the two buds, support the assumption that the distinction between the two buds holds true in the adult pancreas. These differences can serve specific clinically relevant purposes, such as directing the surgeon’s attention towards lymph node stations that appear predestined for lymphatic spread, provided that the cancer’s origin within the pancreas is more precisely defined. Beyond the surgical implications, molecular characterization of the pancreatic anlagen may unveil further distinct differences similar to results regarding the accumulation of pancreatic polypeptide. This could lead to molecular differences, which might be therapeutically useful and serve as targets for tailored drug treatment.

Some limitations of the study must be acknowledged. Manual dissection is prone to investigator bias. Therefore, it is of paramount importance to develop objective methods for confirmation. Visible quality control was performed by immunohistochemical staining and algorithmic validation. Although the number of cases was still low, the objective of providing a proof of concept was considered successful. All histological sections of the pancreatic head that have not yet been shown are provided in Appendix A.

In conclusion, it is not only possible but also necessary to further study the pancreatic anlagen separately. The pancreas should not be viewed as one organ but rather as two distinct entities that form two independent morphogenetic units, depending on their embryological origins. The units are merged merely topographically into a single organ. Dissection along the embryological fusion plane appears to be a viable procedure for patients and is consistent with case reports published over the last 30 years [21,22,23,24,25,26,27,28]. Innovations in optical technologies may facilitate the visualization of connective tissue, and even differences in pancreatic tissue, which will enable intraoperative discrimination in the future [37]. The need for technical assistance appears to be warranted, given that this study shows how tumorous tissue, for which this method should ultimately be developed, interfered with the topographic anatomy of one of the presented cases (Appendix A). The visualization of landmarks may also be facilitated by an interdisciplinary effort, such as endoscopic identification of the common bile duct and main pancreatic duct during the surgical procedure, similar to a rendezvous approach.

This study provides evidence supporting a possible reorientation towards a differentiated approach, depending on the embryological origin and spatial anatomy of the pancreatic tissue. This rationale may play a pivotal role in the therapeutic management of pancreatic cancer in the future.

## Figures and Tables

**Figure 1 biomedicines-13-01318-f001:**
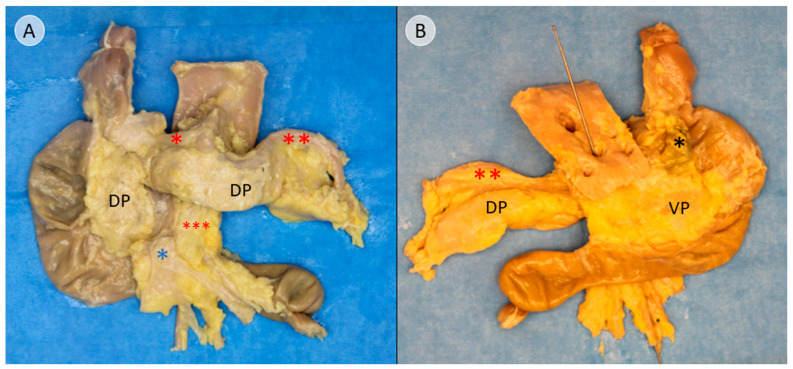
Exemplary pancreatic specimen after extraction with the adjacent duodenum (donor 11). (**A**) shows the ventral view of the pancreas. The ventral surface of the pancreas in the head, as well as in the tail, of the pancreas consists of tissue derived from the dorsal pancreatic bud (DP). The neighboring arterial vessels are marked with asterisks. Common hepatic artery (* in red) and the splenic artery (** in red) both branch off from the celiac trunk. The superior mesenteric artery is also marked (*** in red), as well as the portal vein (* in blue). (**B**) shows the dorsal view of the pancreas. The tissue derived from the ventral pancreatic bud (VP) is limited to the dorsal part of the pancreatic head. The common bile duct has been marked with an asterisk (* in black). The split aorta shows the ostia of the branching vessels. The probe has been inserted into the superior mesenteric artery, which branches off below the celiac trunk.

**Figure 2 biomedicines-13-01318-f002:**
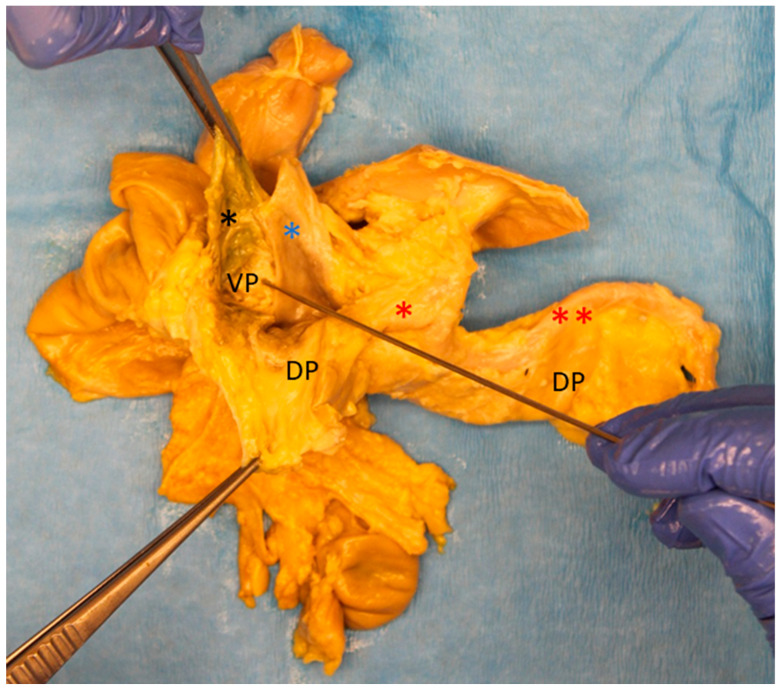
Ventral view of the pancreatic specimen after subtotal separation of the anlagen (donor 11). After dissection along the ventral surface of the common bile duct (* in black) and transection of the main pancreatic duct to facilitate further dissection along the embryological fusion plane, the depicted stage is achieved. The tissue derived from the dorsal pancreatic bud (DP) is folded downward, revealing the tissue that was formed by the ventral pancreatic bud (VP), which is located dorsally to the common bile duct. Folding the DP downward also allows a view on the portal vein (* in blue). The common hepatic artery (* in red) and splenic artery (** in red) are visible along the upper edge of the pancreas after branching off from the celiac trunk.

**Figure 3 biomedicines-13-01318-f003:**
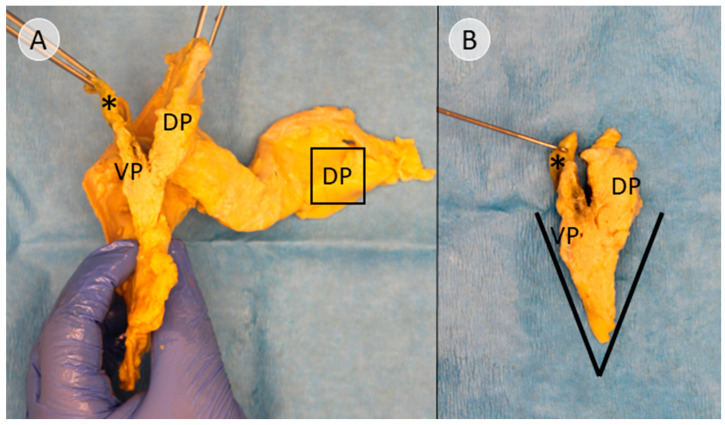
View of the cross-section after sagittal transection of the pancreatic head (donor 11). After subtotal separation of the pancreatic anlagen along the ventral surface of the common bile duct (* in black) and sagittal transection, (**A**) presents a V-shaped cross-section in the remaining pancreatic head. The left leg of the “V” is formed by the dorsally situated tissue that derived from the ventral pancreatic bud (VP), while the right leg of the “V” is formed by the ventrally located derivatives of the dorsal pancreatic bud (DP). (**B**) shows the final resulting specimen after another sagittal incision about 2 centimeters further to the left. The square in (**A**) shows the area of the DP at the far end of the pancreatic tail, from which the negative control for the immunohistochemical staining was taken.

**Figure 4 biomedicines-13-01318-f004:**
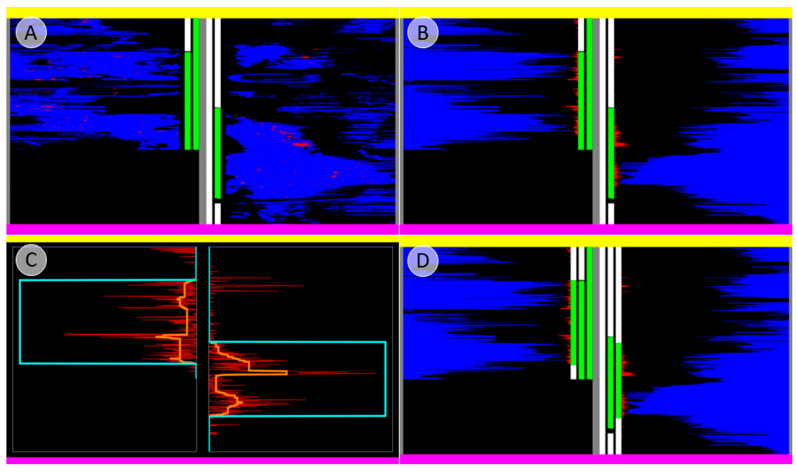
Juxtaposition of the macroscopic tissue designation and the PP-immunostaining (donor 6; see Appendix A for immunostaining). (**A**) The tissue-pixels and PP-pixels along each virtual incision were extracted and depicted in a graph, layer by layer. The layers are sorted from pink (starting point of dissection) to yellow (end point of dissection). The layers belonging to the tissue, which was macroscopically designated to be derived from the ventral pancreatic bud (VP), were layered on the left, with layers from the assumed dorsal pancreatic bud (DP) on the right. This order is represented by the most central opposing bars, with green representing the VP and white the DP. The outer bars represent the manual delineation of the tissue edges based on the PP amount (green: PP-rich; white: PP-poor). In this case, separation was evidently not successful. (**B**) Red and blue pixels were sorted. (**C**) Pixels were counted for each layer for further analysis. The number of red pixels per layer was smoothed, creating the orange line. Data were thresholded via a minimum error global thresholding method, creating the cyan line, which indicates an affiliation with the PP-rich area. (**D**) The third bar for each side represents the tissue affiliation based on the algorithm in (**C**) and the cyan line (green: PP-rich; white: PP-poor).

**Figure 5 biomedicines-13-01318-f005:**
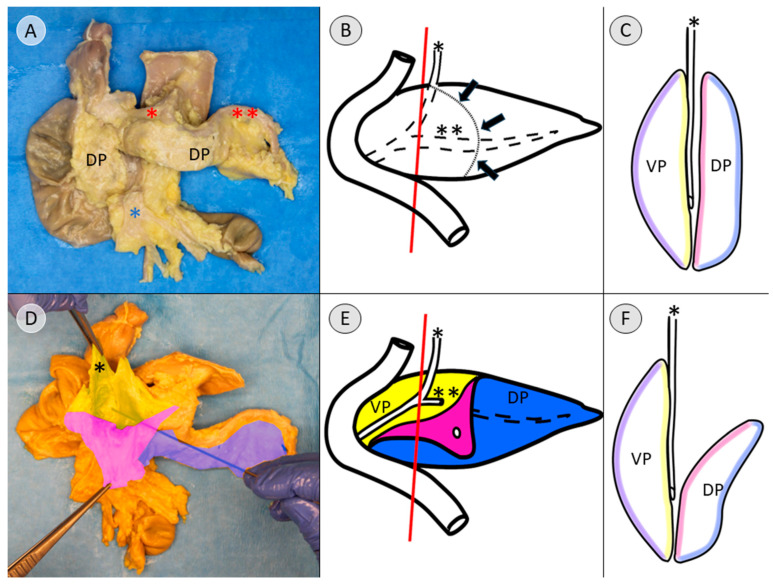
Identifying the developmentally defined boundaries of the pancreatic compartments after manual dissection and separation of the pancreatic anlagen along the embryological fusion plane (donor 11). (**A**) The starting position for the pancreatic specimen was already presented in Figure 1A. (**B**) shows a schematic depiction of the anatomy, putting emphasis on the fact that the derivative of the ventral pancreatic bud (VP, outlines indicated by arrows) is not seen from the ventral perspective and is instead covered by the tissue that was formed by the dorsal pancreatic bud (DP). A view of the cross-section after a sagittal transection along the red line would lead to (**C**). Here, the course of the connective tissue serving as boundaries is indicated by specific colors. The compartments derived from the pancreatic buds are still adhering along the embryological fusion plane (yellow). The separation of the pancreas along the embryological fusion plane, as indicated in Figure (**D**), allows for a view of the planes covered by the connective tissue (see Figure 2 for uncolored anatomical image). (**E**) is a further schematic of (**D**), presenting the common bile duct (* in black) and the transected main pancreatic duct (** in black) that were used as landmarks. A view of the cross-section after a sagittal transection along the red line would lead to (**F**), indicating the separation of the pancreatic buds along the embryological fusion plane. Common bile duct = * in black; main pancreatic duct = ** in black; common hepatic artery = * in red; splenic artery = ** in red; portal vein = * in blue; fascia of Treitz = purple; fascia of Toldt = pink; embryological fusion plane = yellow; unnamed prepancreatic fascia= blue; tissue derived from the ventral pancreatic bud= VP; tissue derived from the dorsal pancreatic bud = DP.

**Figure 6 biomedicines-13-01318-f006:**
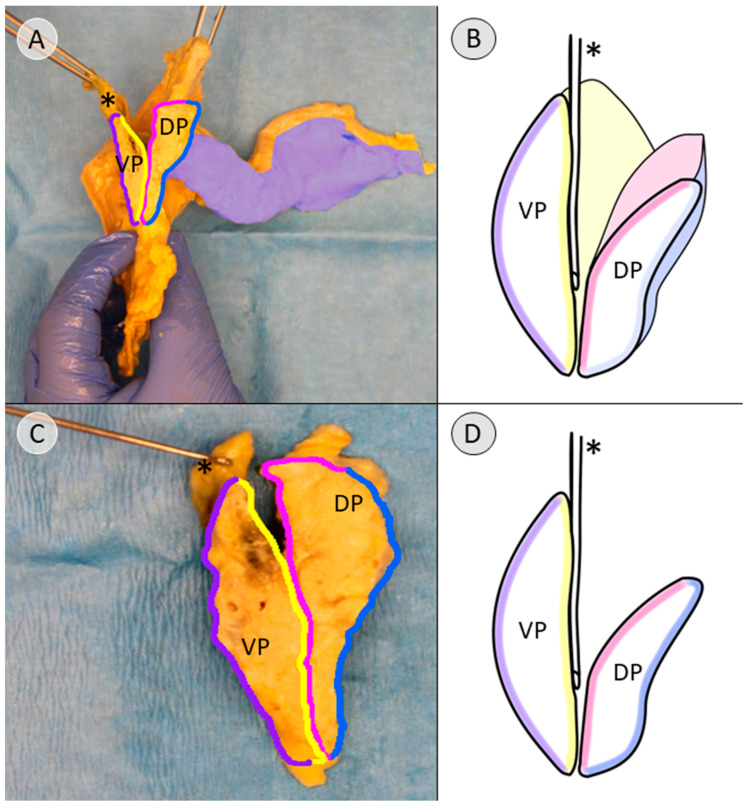
Identifying the developmentally defined boundaries of the pancreatic compartments in the dissected specimen before embedding (donor 11). (**A**) This stage of dissection was previously presented in Figure 3A. The colors indicate the course of the connective tissue serving as boundaries, as explained in Figure 5. (**B**) shows a schematic representation of the anatomy in (**A**) to highlight the course of the connective tissue. (**C**) shows the specimen for histological examination. The expected topography of the tissue derived from the ventral (VP) and dorsal pancreatic bud (DP) is shown, which will be confirmed by immunohistochemical staining. (**D**) depicts a schematic of the assumed anatomy in (**C**). Common bile duct = * in black; fascia of Treitz = purple; fascia of Toldt = pink; embryological fusion plane = yellow; unnamed prepancreatic fascia = blue; tissue derived from the ventral pancreatic bud = VP; tissue derived from the dorsal pancreatic bud = DP.

**Figure 7 biomedicines-13-01318-f007:**
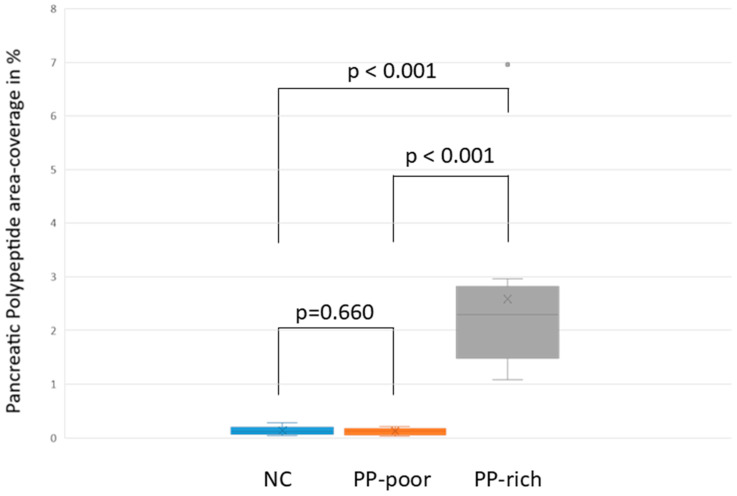
Boxplot comparing the PP coverage of the respective tissue areas in the immunohistochemical staining. The boxplot compares the PP coverage in the negative control (NC; extraction demonstrated in Figure 3) and the PP-poor and PP-rich area in the cross-section of the pancreatic head (extraction demonstrated in Figure 3). The y-axis shows the PP coverage as a percentage. The graph indicates the respective means with an “x”, and the median is indicated by the line within the box. The graph indicates no significant difference (*p* = 0.660) between the NC and the PP-poor area in the pancreatic head. It also indicates a significant difference between the PP-rich area and the NC, as well as the PP-poor area (*p* < 0.001). The graph also shows an outlier in the PP-rich area, with a 6.96% PP coverage, stemming from donor 5.

**Figure 8 biomedicines-13-01318-f008:**
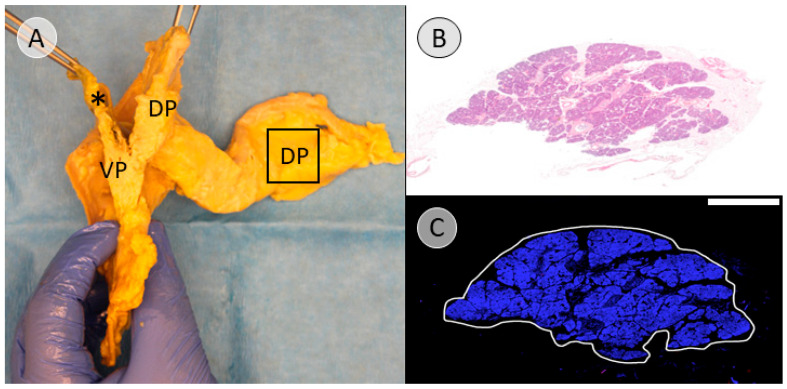
The negative control for the immunohistochemical staining for PP. (**A**) Shown previously in Figure 3A and Figure 6A (donor 11), this figure highlights the location from which the tissue for the negative control will be extracted (indicated by a black square). The location is assumed to be a clear derivative of the dorsal pancreatic bud (DP), which should display a very low accumulation of PP, making it a viable negative control. (**B**) shows a hematoxylin and eosin staining (H&E staining) of the extracted specimen (donor 7). (**C**) reveals the overlay of the DAPI staining and the immunohistochemical staining visualizing PP. The PP staining shows very scarce red spots in the tissue of interest, which is outlined in white for further analysis. The scale bar represents 5 mm. Common bile duct = * in black; tissue derived from the ventral pancreatic bud = VP; tissue derived from the dorsal pancreatic bud = DP.

**Figure 9 biomedicines-13-01318-f009:**
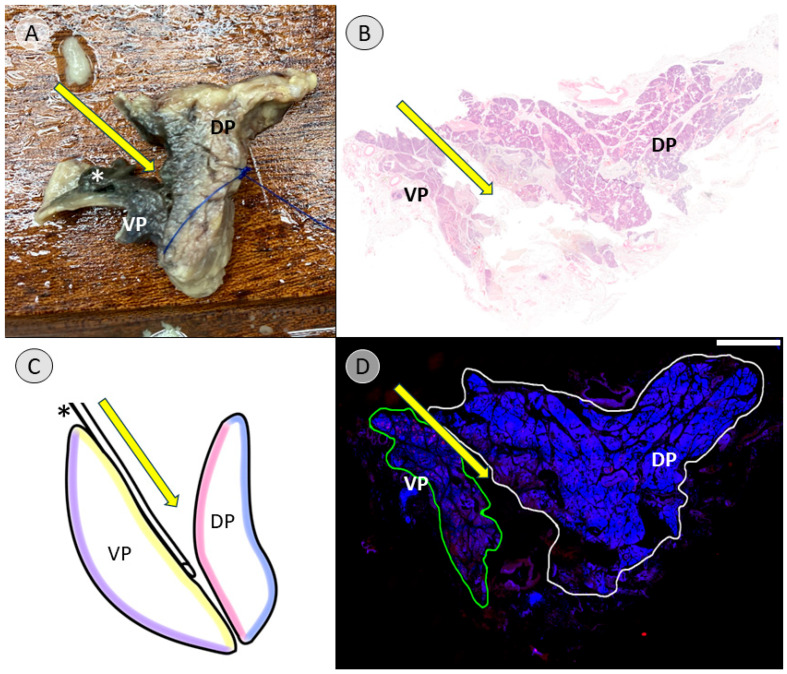
Overlay of the macroscopic, schematic, and histological view of the cross-section of the pancreatic head (donor 7). (**A**) shows the specimen that will be embedded for histological examination, as shown in Figure 3B, with the exemplary specimen. The direction of the manual dissection to separate the tissue derived from the ventral (VP) and dorsal pancreatic bud (DP) is indicated with a yellow arrow, following the plane in front of the common bile duct (* in white). The blue thread indicates the ventral surface of the specimen. (**B**) shows a hematoxylin and eosin staining (H&E staining) of the cross-section of the specimen. The yellow arrow indicates the dissection plane forming an artificial gap within the pancreatic head by manual dissection. VP and DP mark the expected location of the respective compartments. (**C**) shows a schematic representation of the anatomy, with the colored lines indicating the course of the connective tissue serving as boundaries and the yellow arrow showing the dissection plane. (**D**) reveals the overlay of the DAPI (blue) and the immunohistochemical staining visualizing PP (red). The red spots are concentrated in the VP, which is therefore outlined in green for further analysis. The DP outlined in white shows a far lower concentration of red spots. The yellow arrow indicates the direction of dissection. The scale bar represents 5 mm. Common bile duct = * in black/white; fascia of Treitz = purple; fascia of Toldt = pink; embryological fusion plane = yellow; unnamed prepancreatic fascia = blue; tissue derived from the ventral pancreatic bud = VP; tissue derived from the dorsal pancreatic bud = DP.

**Figure 10 biomedicines-13-01318-f010:**
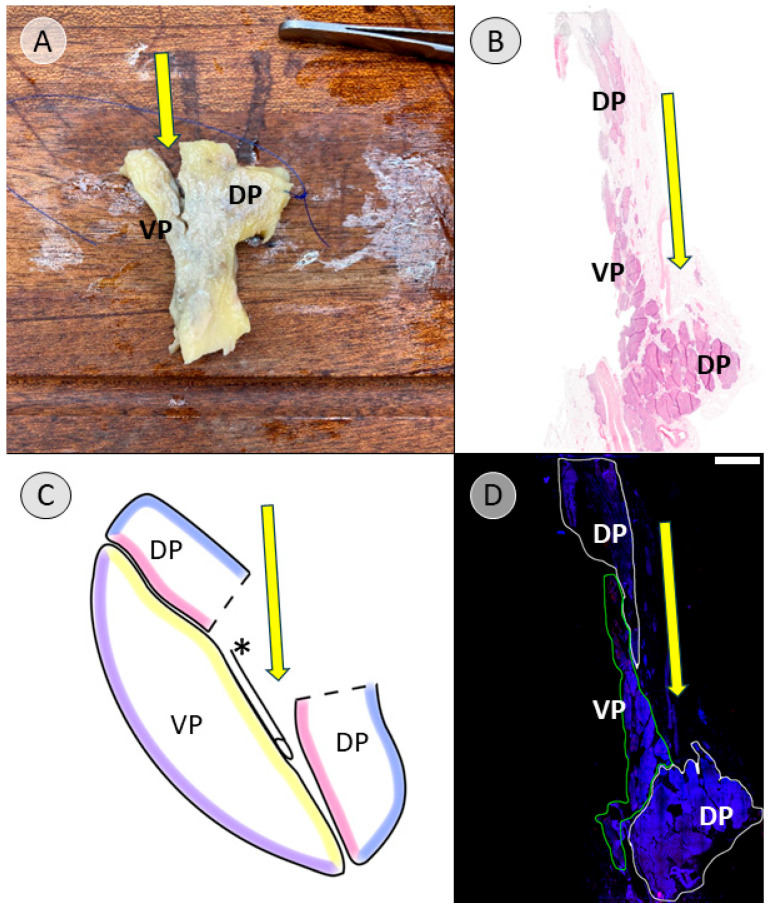
Overlay of the macroscopic, schematic, and histological view of the cross-section of the pancreatic head in a failed separation attempt (donor 2). (**A**) shows the specimen that will be embedded for histological examination, as shown in Figure 3B, with the exemplary specimen. The direction of the manual dissection in an attempt to separate the tissue derived from the ventral (VP) and dorsal pancreatic bud (DP) is indicated with a yellow arrow. The blue thread indicates the ventral surface of the specimen. (**B**) shows a hematoxylin and eosin staining (H&E staining) of the cross-section of the specimen. The yellow arrow indicates the dissection plane and a clear artificial gap caused by manual dissection. VP and DP mark the location of the respective compartments retrospectively after judging the immunohistochemical staining. (**C**) shows a schematic representation of the anatomy, indicating the course of the connective tissue, with the colored lines serving as boundaries and a yellow arrow indicating the direction of the failed dissection plane. (**D**) reveals the overlay of the DAPI (blue) and the immunohistochemical staining visualizing PP (red). The red spots are concentrated in the VP, which is therefore outlined in green for further analysis. The DP outlined in white shows a far lower concentration of red spots. The staining reveals the failure to follow the dissection plane between the anlagen and shows a split DP. The yellow arrow indicates the direction of dissection. The scale bar represents 5 mm. Common bile duct = * in black; fascia of Treitz = purple; fascia of Toldt = pink; embryological fusion plane = yellow; unnamed prepancreatic fascia = blue; tissue derived from the ventral pancreatic bud = VP; tissue derived from the dorsal pancreatic bud = DP.

**Figure 11 biomedicines-13-01318-f011:**
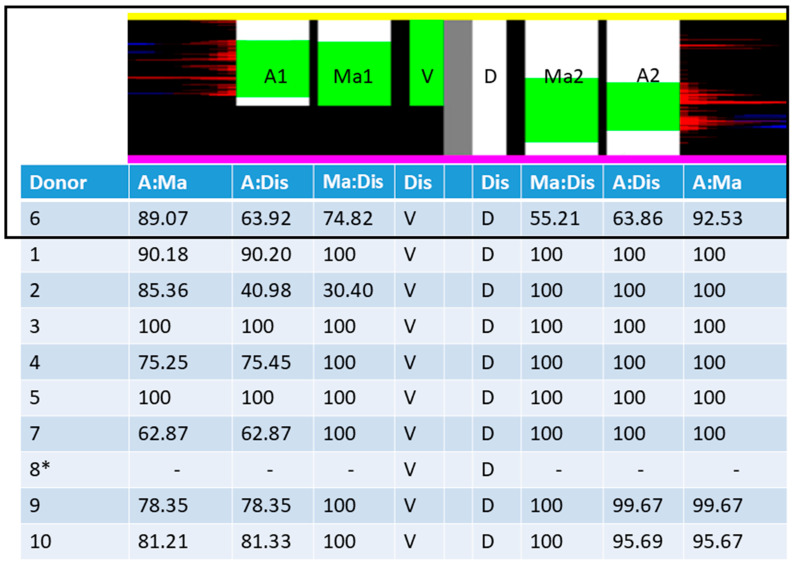
Concordance ratios between the different means of tissue designation along the dissection line to quantify the success of dividing the pancreatic anlagen. The exemplary bars at the top correspond to Figure 4D (donor 6). To emphasize the direction of dissection, the starting point of the dissection has been marked with a pink line, the end point of dissection line has been marked with a yellow line. The tissue layers of the ventral (V) and dorsal (D) pancreatic bud are sorted to the left and right half, respectively, as indicated by the central bars (V and D). This designation was established during dissection (Dis). The adjacent bars, marked A1/Ma1 and A2/Ma2, contain the information regarding how the tissue was identified, either algorithmically (A1 and A2) or by manual delineation, based on the visual PP amount (Ma1 and Ma2). PP-rich areas are coded green and PP-poor areas white. The columns beneath the bars contain the information about the concordance of the algorithmic or manual designation with the designation by dissection (A:Dis and Ma:Dis). Additionally, the concordance ratio between algorithmic and manual designation is given (A:Ma). The asterisk (*) marks the tumorous specimen, in which this compartmentalization could not be achieved. Concordance ratios are given in a percentage and compare the conformity of the depicted bars.

## Data Availability

The original contributions presented in this study are included in the article und Appendix A. Further inquiries can be directed to the corresponding author. The raw data supporting the conclusions of this article will be made available by the authors on request.

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
