# Peer review of "Unraveling the Pancreatic Anlagen: Validating a Manual Dissection Protocol with Immunohistochemical Staining for Pancreatic Polypeptide in a Human Cadaver Study"

_biomedicines, 2025, doi:10.3390/biomedicines13061318_

Round 1
Reviewer 1 Report
Comments and Suggestions for Authors
Your study is one of the best examples of a systematic demonstration of anatomical differentiation of the pancreas based on embryologic anlage both macroscopically and immunohistochemically. Especially the detailed dissection of human cadavers and the successful application of the specific staining protocol with PP make this study remarkable in the literature.
I have some constructive suggestions for this study.
1. Although the introduction is based on a strong literature review, a more neutral and objective language in some sentences (such as “we advocate”, “we are convinced”) would be more appropriate in terms of scientific discourse.
2. The methods section is quite detailed, but the algorithmic analysis process is too technical. This section could be improved by more clearly summarizing the limitations of the algorithms and potential sources of error. Remember the method should be reproducible for every anatomist.
3. The use of language is generally clear, but readability would be improved if some long and complex sentences were simplified. In English, scientific clarity is sometimes sacrificed for fluency.
4. Clinical implications, especially the discussions on Höckel's concept of “morphogenetic unit” are important. However, claims such as “surgical paradigm shift” should be presented with a little more care and restraint given the sample size of the study.
Overall, the study is strong, original and scientific. A more neutral narrative and more clarity on the algorithmic analysis process would further increase the impact of your work.
Comments on the Quality of English LanguageYour text is generally written in clear and technically correct English. However, the preference for excessively long structures in some sentences may make it difficult for the reader to follow. In particular, expressions such as “we advocate” and “we are convinced” should be changed to a more neutral language in scientific writing. In addition, some technical explanations contain very dense information; if these sections are simplified, both fluency and comprehensibility will increase.
Author Response
Dear Reviewer,
Thank you for your kind words and insightful comments.
Our goal was to not leave any room for interpretation in our demonstration of the pancreatic anatomy based on the embryologic anlagen as the nomenclature is not intuitive due to the pancreatic rotation. We are humbled by your supportive words towards our study and are very grateful to be given the chance to contribute something of value to the existing literature.
We would like to point out that the given line numbers and pages in our response are referring to the version of the manuscript in which the track changes are visible.
Comment 1: Although the introduction is based on a strong literature review, a more neutral and objective language in some sentences (such as “we advocate”, “we are convinced”) would be more appropriate in terms of scientific discourse.
Response 1: All personal pronouns have been removed, and a more neutral tone has been adopted (lines 58-59, 63-65, 377-378, 383-385, 385-387 and 392-394). The intention was to clearly indicate instances where our position was being presented; however, it is acknowledged that such phrasing may appear inappropriate within scientific discourse and may compromise the objectivity of the text.
Comment 2: The methods section is quite detailed, but the algorithmic analysis process is too technical. This section could be improved by more clearly summarizing the limitations of the algorithms and potential sources of error. Remember the method should be reproducible for every anatomist.
Response 2: We appreciate the feedback and agree that this section represents a particularly technical aspect of the study. To ensure methodological rigor, we collaborated closely with the data scientist Dr. Karsten Winter, who is a co-author of this work and contributed to the analytical framework. While we acknowledge that the technical depth may pose challenges for reproducibility, we view the interdisciplinary nature of our team as a key strength. The integration of diverse expertise enables the development and validation of solutions that may not be achievable through anatomical insight alone. As you have pointed out, certain sentences in the manuscript are overly complex; we hope that, at a minimum, this section now benefits from clearer and more concise phrasing (line 167, line 191 and line 196).
Comment 3: The use of language is generally clear, but readability would be improved if some long and complex sentences were simplified. In English, scientific clarity is sometimes sacrificed for fluency.
Response 3: We have tried to rephrase long and complex sentences and divide them into more comprehensible and shorter parts (e.g. line 167, line 191 and line 196).
Comment 4: Clinical implications, especially the discussions on Höckel's concept of “morphogenetic unit” are important. However, claims such as “surgical paradigm shift” should be presented with a little more care and restraint given the sample size of the study.
Response 4: We acknowledge that our sample size does not justify the use of the term “paradigm shift” and that this wording may have been premature. The phrase has been revised accordingly, and more measured language has been adopted (lines 392-396).
Comment 5: Comments on the Quality of English Language
Your text is generally written in clear and technically correct English. However, the preference for excessively long structures in some sentences may make it difficult for the reader to follow. In particular, expressions such as “we advocate” and “we are convinced” should be changed to a more neutral language in scientific writing. In addition, some technical explanations contain very dense information; if these sections are simplified, both fluency and comprehensibility will increase.
Response 6: As previously noted, all personal and possessive pronouns have been replaced with professional and neutral language. In addition, greater restraint has been exercised in the formulation of claims, in accordance with the feedback received. Efforts have also been made to simplify overly complex or lengthy sentences, with the aim of enhancing the clarity and accessibility of the technical content.
Reviewer 2 Report
Comments and Suggestions for Authors
Manuscript of great surgical interest supported by the exhaustive study of the anatomy of the pancreas and its ontogenesis.
I would only recommend you to check some errors in the font size of the text as in point 2.3, at the beginning and at the end of this paragraph, on pages 3 and 4.
Author Response
Dear Reviewer,
Thank you for your encouraging verdict considering our work. We were excited to read that you approved of the presented anatomy of the pancreas and its ontogenesis, which forms the core of our hypothesis.
We would like to point out that the given line numbers and pages in our response are referring to the newly uploaded version of the manuscript in which the track changes are visible.
Comment 1: I would only recommend you to check some errors in the font size of the text as in point 2.3, at the beginning and at the end of this paragraph, on pages 3 and 4.
Response 1: Thank you for pointing out the errors in formatting. We have rechecked the manuscript for similar inaccuracies and have resolved the issue in lines 145 and 157.
Reviewer 3 Report
Comments and Suggestions for Authors
The manuscript entitled "Unraveling the Pancreatic Anlagen: Validating a Manual Dissection Protocol with Immunohistochemical Staining for Pancreatic Polypeptide in Human Cadaver Study" by Alvanos and colleagues addresses a crucial medical issue with a very high mortality rate in humans. Surgical intervention is the only possible solution that is currently available, and on which the current manuscript mainly focused upon and suggested some interesting approaches based on the developmental origins of the pancreatic tissue. The developmental origin of the pancreatic tissue was not considered in surgical approaches, which are currently practiced for pancreatic cancer. The authors skillfully approached the pancreatic ontogeny while considering the involvement of the exocrine and endocrine portions in the emergence of pancreatic cancer. My specific comments are as follows:
- Line 42: The word surgical therapy should be changed to surgical approach.
- Line 44: The sentence should be rephrased as "revolutionary in the surgical treatment of cancer".
- Line 46-48: The section should be rephrased as " A surgical approach for cervical cancer, considering an ontogenetic approach to isolate the specific area responsible for tumorigenesis, has been pioneered by Höckel and colleagues.
- Line 48-49: The section should be rephrased as "Interestingly, these specific tumorigenic compartments happened to have common embryological units, hence termed as morphogenetic units".
- Line 54: "in our view" should be removed.
- Line 57-59: The sentence should be rephrased as "It appears to be an oversight to conceptualize the pancreas as a single unit stemming from a single anlagen rather than two independent yet fused morphogenetic units for potential therapeutic or surgical approaches".
- Line 61- 64: The sentence should be rephrased to "In the current study, we aimed to manually dissect along the fusion plane, validated by immunohistochemical staining for pancreatic polypeptide (PP), a valid marker of ventral pancreatic bud-derived pancreatic tissue.
- Line 66-67: The sentence should be rephrased to "We aimed to provide quantifiable evidence for the viability of this dissection approach in terms of consistent results, which are extremely important in surgical oncology".
- Line 133-134: Font size and style for 'Immunohistochemical staining' should be corrected.
- Line 145: Font size and style for Fluorescence mounting medium should be corrected.
- Section 2.4 Digitization and analysis of Immunostained tissue section: As this portion is part of the M&M, the sepcific figures and their numbers should not be mentioned, as these figures are results so they must only be mentioned in detail in the result section.
- Fig 8 C, Fig 9 D, Fig 10 D. The description says "overlay of DAPI (blue) and the IHC staining visualizing PP (red). There are no red areas or points visible in these figures to specify PP localization. The authors can use arrows/arrow heads to pinpoint PP-positive red-stained areas, which the reviewer is unable to locate. Although the original microscopy images provided as a separate file contain elegant red-stained areas. Why did the authors not choose to also add these IHC figures in the manuscript instead of a separate file? I suggest adding these original IHC figures to the manuscript.
- Line 294: The line should be rephrased and should not be a separate line. "The results presented above have promised clinical significance for several reasons discussed below".
Author Response
Dear Reviewer,
We sincerely thank you for the encouraging comments. We appreciate the recognition of our effort to integrate developmental anatomy into the context of surgical approaches for pancreatic cancer. The emphasis on the developmental origins of pancreatic tissue was central to our study, and we are grateful that this aspect was acknowledged.
We would like to point out that the given line numbers and pages in our response are referring to the newly uploaded version of the manuscript in which the track changes are visible.
Comment 1: Line 42: The word surgical therapy should be changed to surgical approach.
Response 1: The wording has been changed as suggested (line 42).
Comment 2: Line 44: The sentence should be rephrased as "revolutionary in the surgical treatment of cancer".
Response 2: The wording has been changed as suggested (line 44).
Comment 3: Line 46-48: The section should be rephrased as " A surgical approach for cervical cancer, considering an ontogenetic approach to isolate the specific area responsible for tumorigenesis, has been pioneered by Höckel and colleagues.
Response 3: We have adapted to your suggestion but changed the phrasing to imply the area in which the cancer cells spread rather than the area in which they originate in terms of tumorigenesis (lines 46-48). We consider this to be a very important point, since the surgical approach can only react to the already formed tumor rather than prevent the tumorigenesis altogether.
Comment 4: Line 48-49: The section should be rephrased as "Interestingly, these specific tumorigenic compartments happened to have common embryological units, hence termed as morphogenetic units".
Response 4: We have implemented your rephrased term but have taken the liberty to leave out “tumorigenic” due to the aforementioned reasons (lines 51-53).
Comment 5: Line 54: "in our view" should be removed.
Response 5: All personal and possessive pronouns have been removed throughout the manuscript, and a more neutral tone has been adopted (lines 58-59, 63-65, 377-378, 383-385, 385-387 and 392-394).
Comment 6: Line 57-59: The sentence should be rephrased as "It appears to be an oversight to conceptualize the pancreas as a single unit stemming from a single anlagen rather than two independent yet fused morphogenetic units for potential therapeutic or surgical approaches".
Response 6: The wording has been changed as suggested (lines 63-65).
Comment 7: Line 61- 64: The sentence should be rephrased to "In the current study, we aimed to manually dissect along the fusion plane, validated by immunohistochemical staining for pancreatic polypeptide (PP), a valid marker of ventral pancreatic bud-derived pancreatic tissue.
Response 7: The wording has been changed as suggested (lines 67-70). However, the personal pronoun was substituted, since it was rightfully critizised as informal and unfitting in an objective setting by one of the co-reviewers.
Comment 8: Line 66-67: The sentence should be rephrased to "We aimed to provide quantifiable evidence for the viability of this dissection approach in terms of consistent results, which are extremely important in surgical oncology".
Response 8: The wording has been changed as suggested (lines 74-76).
Comment 9: Line 133-134: Font size and style for 'Immunohistochemical staining' should be corrected.
Response 9: The font size and style has been corrected (line 145).
Comment 10: Line 145: Font size and style for Fluorescence mounting medium should be corrected.
Response 10: The font size and style has been corrected (line 157).
Comment 11: Section 2.4 Digitization and analysis of Immunostained tissue section: As this portion is part of the M&M, the specific figures and their numbers should not be mentioned, as these figures are results so they must only be mentioned in detail in the result section.
Response 11: We do agree that the figures have the characteristics that fit to the result section. However, as your co-reviewer has pointed out, the section regarding the digitization is very technical. We had added the figures not to present their specific results but rather to guide the reader step-by-step through the process of analysis to enable a better understanding of the resulting figures. The figures in section 2.4 are meant as exemplary figures showing the process with the samples of donor 6. It is essential for the reader to understand the process that is ultimately applied to all 10 donors in order to understand the final Figure 11.
Comment 12: Fig 8 C, Fig 9 D, Fig 10 D. The description says "overlay of DAPI (blue) and the IHC staining visualizing PP (red). There are no red areas or points visible in these figures to specify PP localization. The authors can use arrows/arrow heads to pinpoint PP-positive red-stained areas, which the reviewer is unable to locate. Although the original microscopy images provided as a separate file contain elegant red-stained areas. Why did the authors not choose to also add these IHC figures in the manuscript instead of a separate file? I suggest adding these original IHC figures to the manuscript.
Response 12: We appreciate your feedback regarding our immunohistochemical staining. Regarding Figure 8C, it is supposed to be the negative control for the immunohistochemical staining. PP-positive areas are very scarce and therefore not easy to locate due to the very faint accumulation of PP. However, we can relate to the fact, that the accumulation of PP can be recognized a lot easier in the original microscopy images. Our choice to use the images where the contrast has been digitally adjusted stems from the intention to standardize the images and make them comparable. For the qualitative evaluation however, the original images are far superior. Therefore, we have followed your suggestion and added supplemental figures S7, S8 and S9 that show the histological sections of Figure 8C, 9D and 10D respectively before the contrast was digitally adjusted. We also added the respective histological section into Figure S6. We have referred to these supplemental figures in the text (lines 253-255) and have edited the figure legends accordingly (lines 588-606). The figures S7, S8 and S9 and the updated version of S6 have been uploaded. Also, the former Figure S7 has been changed to Figure S10 to keep the chronological order of the figures.
Comment 13: Line 294: The line should be rephrased and should not be a separate line. "The results presented above have promised clinical significance for several reasons discussed below".
Response 13: The wording has been changed as suggested but we have adjusted the tense to present tense (lines 307-308).